# First Results on the Presence of Mycotoxins in the Liver of Pregnant Fallow Deer (*Dama dama*) Hinds and Fetuses

**DOI:** 10.3390/ani14071039

**Published:** 2024-03-28

**Authors:** István Lakatos, Bianka Babarczi, Zsófia Molnár, Arnold Tóth, Gabriella Skoda, Győző F. Horváth, Adrienn Horváth, Dániel Tóth, Farkas Sükösd, László Szemethy, Zsuzsanna Szőke

**Affiliations:** 1Department of Regional Game Management, Ministry of Agriculture, 1052 Budapest, Hungary; sotakalbulldogs@gmail.com; 2Agribiotechnology and Precision Breeding for Food Security National Laboratory, Institute of Genetics and Biotechnology, Department of Animal Biotechnology, Hungarian University of Agriculture and Life Sciences, 2100 Gödöllő, Hungary; babarczi.bianka@uni-mate.hu (B.B.); molnar.zsofia@uni-mate.hu (Z.M.); toth.arnold@uni-mate.hu (A.T.); skoda.gabriella@uni-mate.hu (G.S.); ferenczine.szoke.zsuzsanna@uni-mate.hu (Z.S.); 3Institute of Biology, University of Pécs, 7624 Pécs, Hungary; hgypte@gamma.ttk.pte.hu (G.F.H.); horvath.adrienn.1989@gmail.com (A.H.); tothdaniel0@gmail.com (D.T.); 4Institute of Pathology, University of Szeged, 6720 Szeged, Hungary; sukosd.farkas@gmail.com

**Keywords:** Zearalenone, Aflatoxin, DON, T2-toxin, Fumonisin B1, fallow deer, hind, fetus

## Abstract

**Simple Summary:**

Mycotoxins, which are secondary metabolites of fungi, have several adverse effects on both animal and human organisms. Because of climate change, mycotoxins are gradually spreading from the south to the north and west in Europe, causing a growing risk for feed and food. The toxic influences of mycotoxins are intensively studied in domestic animals, but much less is known about the game species. Reproductive abnormalities have been observed in a valuable game species, the fallow deer. We supposed mycotoxin contamination to be one of the possible causes because it is known to be dangerous even at low toxin levels, especially for young animals. As the mycotoxin exposure of the fallow deer living in a forest–agriculture complex habitat is not known, our goals were (i) to determine the mycotoxin presence in hinds; (ii) to detect the spatial and individual variability in mycotoxin levels; and (iii) to reveal the mycotoxin concentration in the fetuses in relation to their mothers. A total of 72 fallow deer embryos and their mothers were sampled in seven forested regions in Hungary in the 2020/2021 hunting season. Mycotoxin metabolites were analyzed by ELISA from the liver tissue. We detected the studied mycotoxins (Aflatoxin, Zearalenone, Fumonisin B1, DON, T2/HT2-toxin) in 41–96% of fallow deer hinds and 3–98% of the fetuses. All mycotoxins were passed into the fetus, but only Fumonisin B1 rarely passed. The individual variability of mycotoxin levels was extremely high, which obscured the spatial differences among the study sites. We concluded a possible accumulation of ZEA and DON in the fetuses because of the higher concentrations in the fetus than in the hind. These results reflect the potential threats of mycotoxins to the population dynamics and reproduction of wild fallow deer.

**Abstract:**

Reproductive abnormalities have been observed in fallow deer populations in Hungary. We supposed mycotoxin contamination to be one of the possible causes because multi-mycotoxin contamination is known to be dangerous even at low toxin levels, especially for young animals. We investigated the spatial pattern of mycotoxin occurrences and the relationship between maternal and fetal mycotoxin levels. A total of 72 fallow deer embryos and their mothers were sampled in seven forested regions in Hungary in the 2020/2021 hunting season. We analyzed Aflatoxin (AF), Zearalenone (ZEA), Fumonizin B1 (FB1), DON, and T2-toxin concentrations in maternal and fetal livers by ELISA. AF was present in 70% and 82%, ZEA in 41% and 96%, DON in 90% and 98%, T2-toxin in 96% and 85%, and FB1 in 84% and 3% of hind and fetus livers, respectively. All mycotoxins passed into the fetus, but only Fumonizin B1 rarely passed. The individual variability of mycotoxin levels was extremely high, but the spatial differences were moderate. We could not prove a relation between the maternal and fetal mycotoxin concentrations, but we found an accumulation of ZEA and DON in the fetuses. These results reflect the possible threats of mycotoxins to the population dynamics and reproduction of wild fallow deer.

## 1. Introduction

Mycotoxins are secondary metabolites of fungi that have several adverse effects on animals and humans as natural environmental stressors. Mycotoxins can cause a wide range of toxic effects in humans and animals, from carcinogenesis or DNA damage to targeting specific organs such as the kidney, liver, or intestines. Moreover, mycotoxins can cause reproductive problems and harm embryos and fetuses. Furthermore, the co-occurrence of mycotoxins in a single exposure may further increase the potential health risks. In addition, multiple co-occurring and acting agents may interfere in an additive or synergistic manner [1]. Therefore, the detection (and accurate measurement) of all relevant mycotoxins may be of utmost importance, but it is currently not feasible. An appropriate/ideal (or “definitive”) solution should be able to detect selected mycotoxins simultaneously and possibly be dedicated to the local problem [2,3,4]. In addition, the detection of both the exposure and accumulation levels (e.g., in relevant organs or body fluids) for different species would be the most useful. Finding and linking the source (accumulated levels), and ideally, the induced biological effects, are the truly preferred approach [5]. The recommended limits for environmental concentrations of toxins, which are largely based on the results of studies on toxic levels and much less on sub-toxic concentrations, are relevant in studies on the biological effects of mycotoxins, but the duration of exposure is also crucial. However, the implementation of these studies would be necessary to reveal the real harm caused by mycotoxins, but we are far from doing so, especially for wild animals.

Because of climate change, the Aflatoxins (AFs) emerge more frequently in northern and western Europe, causing a growing risk to feed and food [6,7]. In general, the spread of molds that infect plants, associated plant diseases, and consequently, mycotoxin contamination of food and feed supplies are predicted to increase with global warming and climate change [8]. As the influence of climate change becomes stronger in the Carpathian Basin, mold fungus occurrence and mycotoxin contamination have become increasingly serious problems in Hungary [9]. The prevalence of Fusarium toxins is increasing, and new mold species such as *Aspergillus flavus* and new mycotoxins, especially AFs, have emerged in Hungary over the last 15 years [9]. The spatial and temporal dispersion and intensity of mycotoxin emergence are highly variable depending on extreme and changing weather conditions.

More than 18 AFs are currently known, but AF B1 (AFB1) is by far the most studied. They are mutagenic, tumorigenic, and carcinogenic [10]. Exposure of pregnant females to AFs can affect embryo development in utero, causing various adverse health effects and abnormal pregnancy outcomes [11]. In mammals, the systemic circulation of highly exposed mothers can deliver AFs or their toxic metabolites to the fetus, as has been demonstrated in highly exposed pregnant women from some African and Asian countries as well as in animals. Indeed, AFs and/or their derived biomarkers, such as AF metabolites, AF-DNA, and AF-albumin adducts, have been detected in fetal cord blood and in both fetal cords and maternal blood samples [12,13,14]. Accordingly, it has been concluded that AFs and their metabolites pass into the fetus in pregnant women and are metabolized via the same pathway as in adults [15]. The immunosuppressive effects of AFs, including reduced antibody production, increased susceptibility to infectious diseases (such as salmonellosis, candidiasis, coccidiosis, and liver fluke infection), and reduced cell-mediated immunity, have been extensively studied in several animal models [16].

Estrogenic-like Zearalenone (ZEA) and its metabolites are endocrine-disruptor compound (EDC) agents that can damage the hormonal system. ZEA is a non-steroid estrogenic mycotoxin with a structure similar to that of naturally occurring estrogens [17]. ZEA can bind to alpha and beta estrogen receptors (ERs) and interfere with the endocrine system [18]. By pairing with ERs and their structural mimicry of estrogens, myco-estrogens act as negative regulators of gonadotropin-releasing hormone (GnRH) [19]. Their effects include fertility problems in males and females. In pregnant animals, persistent ZEA exposure decreases the survival of the embryo, expected fetal weight, and amount of breastmilk during lactation. In lower levels, ZEA can cause endometrial hyperplasia and therefore, it elevates the risk of endometrial cancer in humans [20,21]. In the liver, which is well known as the main organ of ZEA metabolism, this mycotoxin induces pathological changes and, subsequently, liver cancer [22]. ZEA exerts hematotoxic effects by disrupting blood clotting and altering blood parameters [22].

Deoxynivalenol (DON) is the most commonly detected mycotoxin contaminant in cereal crops and cereal-based food products in temperate regions of the world. DON causes adverse health effects in animals, passes through the fetus, and causes fetal abnormalities. Research on the effects of DON on placental transfer remains fragmentary. The transfer of DON through the placenta to the fetus has been observed in different species. For example, in pregnant sows, DON was detected in fetal plasma [23] and the liver, as well as in the kidney [24], and it was associated with fetal growth restriction [25].

In vitro, DON has direct effects on cells of the reproductive system, for example, on ovarian cells by altering oocyte maturation, and on BeWo (isolated from the placenta of a patient with choriocarcinoma) cells by inhibiting granulosa cell proliferation. DON has been shown to alter fetal growth and cause bone malformation in vivo. Although the roles of Akt (serine/threonine-specific protein kinase) and MAPKs (mitogen-activated protein kinases) have been elucidated in the porcine endometrium, the potential effects of DON on placental function and embryogenesis have not been adequately evaluated [26].

The reproductive toxicity of the T2-toxin is evidenced by decreased fertility, disrupted structures and functions of reproductive organs, and the loss of gametogenesis in males and females. T2-toxin disrupts the reproductive endocrine axis and inhibits reproductive hormone synthesis. Furthermore, exposure to T2-toxin during pregnancy results in embryotoxicity and abnormal offspring development [27].

Fumonisin B1 (FB1) is a mycotoxin produced by the fungus Fusarium verticillioides, a common contaminant of corn worldwide. FB1 disrupts sphingolipid biosynthesis by inhibiting ceramide synthase, resulting in an elevation in the level of free sphingoid bases and the depletion of downstream glycosphingolipids. A relationship between a maternal ingestion of FB1-contaminated corn during early pregnancy and an increased risk of neural tube defects (NTDs) has recently been proposed in human populations worldwide, where corn is a dietary staple [28]. Maternal exposure to FB1 altered sphingolipid metabolism and folate concentration in LM/Bc mice, resulting in a dose-dependent increase in NTDs, which could be prevented by maintaining adequate folate levels. Previous studies [25] have suggested that FB1 does not cross the placenta but acts on embryonic development by causing folate deficiency.

The toxic influences of mycotoxins are intensively studied in domestic animals, especially ruminants [29,30,31,32]. The belief that ruminants are less susceptible to mycotoxins likely stemmed from a historical focus on monogastric animals (like pigs and poultry) and their primary feed sources (cereals) in mycotoxin research [29,33]. This has led to a shift in focus, with several studies investigating the presence of various mycotoxins in ruminant feed components, including pastures, cereals, and silage [34]. The proactive management of dietary levels of toxins and endocrine-disrupting metabolites in cattle is essential to minimize risks to their health, reproduction, and production [35].

Much less is known about game species. Although the fallow deer *(Dama dama)* is an introduced species in the Carpathian Basin, it has become a valuable game species for Hungarian game management because of the continuous increase in population size and harvest, and the expansion in the last decades has provided good-quality trophies, an increasing mass of venison, and considerable incomes [36]. Fallow deer and other large herbivores play important roles in the ecosystem via the regulation of plant species and as important preys of carnivores.

Despite the long-term population increase, game managers have observed reproductive problems and a sharp decrease in the number of fawns throughout the country. As the usual explanations for reproduction problems (predation, poaching, disturbance, etc.) are not satisfactory, more hidden and less-obvious causes must be considered. We suggested that mycotoxin contamination, or more specifically, multi-mycotoxin effects, is a possible cause of decreased reproduction. As the venison is premium food for humans, the potential decrease in harvest and any kind of contamination can cause economic and food safety problems. Fallow deer live in forest–agricultural complex habitats, so they have access to agricultural crops. Game fields and supplementary feed, especially grains provided by game managers, can be mycotoxin sources.

As there are very few scientific results on the mycotoxin sources available for free-living wild animals, the mycotoxin contamination of games, and the specific effects of the mycotoxin on these species and on the ecosystems indirectly, we have started a long-term study to reveal (i) the presence of mycotoxins in games, especially fallow deer; (ii) possible threats to the populations; and (iii) potential mycotoxin sources. As the first step, we analyzed ZEA, AF metabolites, DON, T2-toxin and HT2-toxin, and FB1 mycotoxins to check the possibility of a mycotoxin contamination of pregnant hinds. The goals of our study were: (i) to determine the mycotoxin presence in the hinds, (ii) to detect the spatial and individual variability in mycotoxin levels, and (iii) to reveal the mycotoxin concentration in the fetuses in relation to their mothers.

## 2. Materials and Methods

### 2.1. Study Areas

Sampling was carried out in seven game management units (GMU1: Gyönk; GMU2: Törökkopány; GMU3: Gúth; GMU4: Kocsola; GMU5: Tamási; GMU6: Mészkemence; GMU7: Kelebia) in the forested areas of the Great Hungarian Plain and South Transdanubian Hills (Figure 1).

Based on game management plans, the ecological environment and main management characteristics were similar in the sampling areas. The climate is continental. The yearly average precipitation is 500–650 mm with unequal dispersion; drought occurs frequently, especially in summer. The average annual temperature varies from 9.5 to 11.5 °C. Forests are mainly broad-leaved forests, dominated by oaks (*Quercus robur* and *Q. cerris*) and black locust (*Robinia pseudoacacia*), and surrounded by large agricultural lands where maize, wheat, sunflower, and alfalfa are the main products. Game management was similar in the sampling areas, and the fallow deer was the dominant game species. Trophy hunting is the main goal of fallow deer management; however, venison production is also important. Managers use intensive methods with strict population control, habitat and game field management, and supplementary feeding from autumn to spring to maintain good-quality populations.

### 2.2. Sampling

The hunts were carried out during the regular hunting season of December–January 2020/2021. Although most of the fallow deer hinds were in the first trimester of pregnancy, the pregnant and non-pregnant hinds could not be distinguished in this phase. As our main goal was to study whether the mycotoxin can pass through to the fetus, we selected pregnant hinds for sampling during evisceration regardless of age or condition. The liver and uterus with the developing fetuses were removed within 3 h after shooting and stored frozen until the analysis. Analyses were performed on liver samples from 70 fallow hinds and 72 fetuses, as two hinds had twins. The ages of the fetuses were classified into three groups—1: one-month-old, 2: two-months-old, 3: three-months old—based on body weight and length according to the methods developed for Hungarian fallow deer populations [37,38]. The sample sizes may differ in the analyses because of inappropriate or missing liver samples. In the case of eight hinds, the liver samples were missing or unsuitable for mycotoxin analysis due to serious American liver fluke (*Fascioloides magna*) infection. In some other cases, only a small quantity could be taken, especially from very small fetuses, or the liver of the hind limb was damaged from the shooting. The sample size was not sufficient for the analysis of all the toxins in these cases. The actual sample sizes are presented in Table 1 and Table 2.

### 2.3. Toxin Analyses

Mycotoxin measurements were performed on the liver of both pregnant fallow deer hinds and their embryos. The assays were performed using the ELISA method, preceded by a mycotoxin metabolite assay method optimization for the species and organs. ZEA, FB1, DON, T2/HT2-toxin, and AF metabolites were analyzed by an immunoassay.

The stored liver samples were ground and homogenized using a blender. Aliquots were then weighed into extraction tubes and stored at −20 °C until further analysis.

To measure toxin levels, samples were thawed, and 0.5 g of tissue was homogenized in 0.5 mL of ice-cold 50 mM sodium acetate buffer (pH = 4.80). Homogenization was carried out using a FastPrep-24 Classic (MP Biomedicals, Irvine, CA) homogenizer with metal beads and incubated for 3 h at 37 °C in a shaker with the addition of *Helix pomatia* β-glucuronidase/aryl sulfatase (Roche BGALA-RO), according to the manufacturer’s instructions. After incubation and extraction with a mixture of 70% (*v*/*v*) methanol and water (3:1 *v*/*v*), the samples were shaken for 15 min at room temperature (RT) on an orbital shaker. The extracts were centrifuged (RT: 22–25 °C, 5 min, 8000× *g*), and the supernatants were collected and diluted with 0.01 M PBS at pH7.4. The dilution factor of the supernatant was 6–8× (depending on the toxin).

The total AF and DON contents were determined using Toxi-Watch (Soft Flow Ltd., Pécs, Hungary) and ZEA Ridascreen (R-Biopharm, No.1401, Arnhem, The Netherlands) immunoassays. The limit of detection (LOD) of AF B1 was 0.075 ng/g and that of the DON assay was 1.5 ng/g. In the ZEA Ridascreen test, the LOD was 0.075 ng/g. For T2/HT2-toxin detection, the Bio-Shield T-2/HT-2 (Prognosis Biotech, Larissa, Greece) ELISA test was used, with an LOD of 0.5 ng/g. Fumonisin toxins were measured using the EUROPROXIMA Fumonisin (5121FUM) (R-Biopharm Netherlands BV, Arnhem, The Netherlands) ELISA test. Analyses were performed according to the manufacturer’s instructions, and the LOD value was 1.0 ng/g.

Before the ELISA, we compared 10 liver samples/different mycotoxins with HPLC/FLD using our validation methods [5].

### 2.4. Statistical Analysis

In the case of fallow deer hinds and fetuses, all mycotoxin concentrations in the liver are presented as medians. The range and number of cases in which the concentration was zero are also indicated. (The values of samples in which the given mycotoxin was not detected, or the concentrations were below LOD, were considered zero). After the analysis of normality using the Shapiro–Wilk test [39], we used nonparametric statistics to evaluate the results. Boxplots (median, 25–75% percentiles, min–max values, and distribution of data) were used to determine the distribution of mycotoxin concentrations. In each game management unit, we used a Kruskal-Wallis ANOVA (followed by Dunn’s post-hoc test for multiple comparisons) to test the differences in the mycotoxin levels of hinds and fetuses, and the Mann–Whitney test was used to examine the difference between the toxin level of the hinds and fetuses. We also used the Kruskal–Wallis test to compare mycotoxin concentrations between different fetal age groups [39]. To analyze the hypothetical relationship between mycotoxin levels in fallow deer hinds and their fetuses, generalized additive models (GAMs) were performed in the “mgcv” (ver. 1.7–23) package in R [40]. GAMs are appropriate because they detect linear or nonlinear relationships between a given response and relevant predictor variables. In our case, we considered the mycotoxin concentration of the fetuses as a response variable, while the levels of mycotoxins in the fallow deer hinds and their interactions were the predictors. The GAM procedure automatically selects the degree of smoothing based on the Generalized Cross Validation (GCV) score. GCV is a proxy for the model’s predictive performance, analogous to Akaike’s Information Criterion [41]. All statistical tests were performed using R v. 4.2.3 [42]. Statistical tests were considered significant at *p* ≤ 0.05 in all analyses [43].

## 3. Results

### 3.1. Mycotoxin Levels in the Liver of Fallow Deer Hinds and Fetuses

AF was present in 70%, ZEA in 41%, DON in 90%, T2/HT2-toxin in 96%, and FB1 in 84% of hinds. The mycotoxin concentrations showed high individual variability in the hind livers (Table 1). The minimum concentrations were “0” for every mycotoxin. In the case of ZEA, more than half and, regarding AF, nearly one-third of the hind liver samples did not contain the toxin. In some cases, the maximum concentration was extremely high.

### 3.2. Spatial Patterns of Mycotoxin Concentrations in the Hinds and Fetuses

Comparing the mycotoxin concentrations, we rarely found significant differences among sampling sites.

ZEA concentrations in hind livers were significantly higher in GMU6 than in GMU3 (Kruskal–Wallis test: H (6, N = 59) = 18.526, *p* = 0.005; post-hoc Dunn test: z = 3.109, *p* = 0.040) (Figure 2A).

Mycotoxin levels did not differ in the case of hinds for AF and DON (H (6, N = 60) = 12.414—12.503, n.s.) (Figure 3A and Figure 4A). However, the T2-toxin concentration was significantly different among sampling sites (H (6, N = 53) = 20.765, *p* = 0.002), and the concentration was higher in GMU5 than in GMU2 (post hoc: z = 3.184, *p* = 0.031) and GMU3 (post hoc: z = 3.063, *p* = 0.046) (Figure 5A).

The FB1 concentration levels also differed among the game management units (H (6, N = 59) = 27.324, *p* = 0.0001), and were lower in GMU1 than in GMU2 (post hoc: z = 4.100, *p* = 0.001) and GMU5 (post hoc: z = 3.843, *p* = 0.003), and it was also lower in GMU7 than in GMU2 (post hoc: z = 3.300, *p* = 0.020) and GMU5 (post hoc: z = 3.052, *p* = 0.048) (Figure 6A). The fetal ZEA concentration in the liver was significantly higher in GMU7 than in GMU3 (H (6, N = 72) = 21.155, *p* = 0.002; post hoc: z = 4.211, *p* = 0.005) (Figure 2B). AF concentrations also differed significantly among the sampling sites (H (6, N = 65) = 21.196, *p* = 0.002). The toxin level was significantly higher in GMU4 than in GMU2 (post hoc: z = 3.274, *p* = 0.022), GMU5 (post hoc: z = 3.665, *p* = 0.005), and GMU6 (post hoc: z = 3.378, *p* = 0.015) (Figure 3B). DON concentrations were also significantly different among the game management units (H (6, N = 60) = 29.417, *p* = 0.0001). DON concentration in GMU2 was lower than that in GMU1 (post hoc: z = 3.438, *p* = 0.012) and GMU7 (post hoc: z = 4.318, *p* = 0.0003), and significantly higher toxin concentrations were detected in GMU7 than in GMU3 (post hoc: z = 4.073, *p* = 0.001) (Figure 4B). No significant differences were found in the case of the T2-toxin (H (6, N = 71) = 1.965, n.s.) (Figure 5B). In the case of the FB1 toxin, we did not perform this statistical analysis due to the small sample size (Figure 6B).

### 3.3. Comparison of the Mycotoxin Levels between Fallow Deer Hinds and Fetuses

Considering that the difference in mycotoxin concentrations between the fallow deer hinds and their fetuses was assessed using the Mann–Whitney U test, the ZEA levels were significantly higher in the fetuses than in the hinds in six of the seven sampling areas. AF concentrations were similar, with the only difference observed in the GMU4 area (Table 3). We found an opposite relationship between DON and T2-toxin. Regarding DON, there was a significant difference in the GMU2 area; the toxin level was higher in the hinds than in the fetuses (Table 3). In a few cases, there was also a significant difference in the concentration of T2-toxin; the toxin levels in the fetuses were higher in GMU1 and GMU5 (Table 3).

### 3.4. Difference in Mycotoxin Levels among the Three Age Groups of Fetuses

We tested the difference in mycotoxin levels among the three age groups of fetuses based on a Kruskal–Wallis test. Although a slight increase in mycotoxin concentrations was suspected because the age of fetuses in the cases of ZEA, AF and DON, the differences were not statistically significant (ZEA: H (2, N = 72) = 2.985, n.s.; AF: H (2, N = 65) = 2.025, n.s.), except for DON, where the toxin level of the third, most developed class was significantly higher than in the case of the youngest group (H (2, N = 60) = 6.100, *p* = 0.047; post hoc: z = 2.445, *p* = 0.044) (Figure 7).

### 3.5. Relationships between the Mycotoxin Level of the Hinds and Their Fetuses

Mother–fetus data pairs were divided into three groups for the analysis of the relationship between hind and fetus mycotoxin levels: (1) concentration was “0” in hind for the given mycotoxin but it could be detected in the fetus liver; (2) the given mycotoxin was present in the hind liver, but it was absent from the fetus; and (3) both the hind (mother) and the fetus contained the given mycotoxin. A total of 27 mother-fetus data pairs were classified into Group 1 for ZEA, 11 cases for AF, 6 cases for DON, 8 cases for T2-toxin, and no cases for FB1, so this group was excluded from the preliminary correlation tests. Group 2 and Group 3 data were involved in the correlation analysis.

Considering the GAM modeling, the fitting of the smoothed (non-linear) models were better than the parametric (linear) term regarding all mycotoxins. The predicted non-linear relationship between the hinds and the fetuses in main effect models was significant in the case of ZEA (ZEA_foetus_~s(ZEA_hind_): edf = 4.868, F = 7.740, *p* = 0.006) and AF (AF_foetus_~s(AF_hind_): edf = 5.699, F = 2.671, *p* = 0.043). In the case of the T2-toxin concentration, the T2-toxin×FB1 interaction model was supported as the best candidate model. Although the non-linear interaction effect was significant (T2-toxin_foetus_~s(T2-toxin_hind_, FB1_hind_): edf = 2.603, F = 4.024, *p* = 0.018), the model did not prove that the different FB1 concentrations significantly influenced the positive effect of T2-toxin concentration in hinds.

## 4. Discussion

Mycotoxins contaminate cereal grains and other vegetables worldwide, and their presence in animal feed poses a potential health threat to farm animals (and human consumers). Several studies have been conducted on these subjects. It is known that the mycotoxins studied could have a strong negative influence on pregnant females and embryos. AF can cause various adverse health effects and different abnormal pregnancy outcomes [11]. Estrogenic-like ZEA can damage the hormone system, and persistent ZEA exposure decreases embryo survival [20,21]. DON passes through the fetus and causes abnormalities [23,24,25]. DON exposure during pregnancy alters the mRNA and protein expression of junctional proteins in the placenta [44]. The embryotoxic and teratogenic effects of T2-toxin and its metabolite, HT2-toxin, are not negligible, as they can cause miscarriage in several mammalian species. Exposure to T2-toxin during pregnancy results in embryo toxicity and an abnormal development of the offspring [27]. In cattle, managing the dietary levels of toxins and endocrine-disrupting metabolites is crucial for assessing and mitigating risks to health, reproduction, and production [32,33,34].

Wild animals living in natural environments have been studied less. Knowledge of domestic ruminants is hardly applicable to deer species because of differences in diet, digestion physiology, and metabolism [45]. Wild animals, especially herbivores living in forest–agriculture complex habitats, can feed not only on natural plants but also on cultivated crops. Game managers also provide agricultural products, particularly grains, as supplementary feed for game populations. Consequently, it is a correct presumption that wild ruminants can be affected by mycotoxin contamination, which can also threaten population growth and quality. As Hungarian game managers reported reproductive problems of one of the most valuable game species, the fallow deer, and the frequency of occurrence of mycotoxins increased in the country, we have started a study on the influence of mycotoxins on fallow deer. As we do not know anything about mycotoxin exposure in free-living games, the first step was to prove the presence of mycotoxins in the hinds and in their fetuses. In this paper, we discuss the first descriptive results of the AF, Zearalenon, DON, T2-toxin, and FB1 levels measured in the liver of fallow deer hinds and fetuses.

We successfully detected all the selected mycotoxins in hinds and fetuses. This proves that most of the studied mycotoxins can pass through to some extent into the fetus; however, except for two samples from a single sampling site, we could not confirm the passage of fumonisins, which is in perfect agreement with the literature [46].

The data showed high individual variability in mycotoxin concentrations in the maternal and fetal livers in each study area. Although many individuals were free of mycotoxins, several contained one or more mycotoxins at very high concentrations. Similar results were found in a Polish study of red and roe deer [47]. They analyzed ZEA and α-zearalenon in the blood plasma. The concentrations in red deer hinds and roe deer were much higher than those found in the liver of fallow deer hinds, but similar to the levels in fetuses.

Comparing the sampling sites, we rarely found significant differences in mycotoxin concentrations. The high individual variability obscured spatial differences. This reflects the diversity of habitats in the game management units, the variability of habitat use by individuals, and spatial and temporal changes in the accessibility of mycotoxin resources in every sampling area. These circumstances make the investigation more difficult than in the case of domestic animals, where feeding experiments and physiological measures can be performed more easily. Diet composition and mycotoxin resources were not investigated in this study, but we suppose that supplementary feeding, especially maize, could be one of the most probable sources [48,49], and other sources, such as natural and cultivated plants, can also be contaminated by different mycotoxins [29,30,34]. Further studies are necessary to reveal and filter out the main mycotoxin sources and the dynamics of mycotoxin contamination.

We hypothesized a relationship between maternal and fetal mycotoxin concentrations, but we could not prove that. The spatial patterns of mycotoxin concentrations differed between the hinds and fetuses. The concentration of the mycotoxin did not change in parallel between the two groups at the same site. The GAM modeling—and other regression analyses—did not provide strong and clearly understandable results. We categorized the mother–fetus data pairs into three groups. In Group 1, the hinds were free of mycotoxins, but the fetus contained them. We assumed that the former intake of the given toxin had been eliminated from the hind liver, but was still detectable in the fetus. As we did not know the quantity of former mycotoxin intake, we had to ignore this group in the statistical analysis, but these data reflect a time delay in the elimination of different toxins in the fetus. In Group 2, the given mycotoxin was present in the hind liver, but was absent from the fetus. We assumed that a recent mycotoxin intake by the mother had not yet penetrated the fetus. This could reflect the possible filter capacity of the placenta or that infiltration into the fetus requires a longer time. If both the maternal and fetal liver contained the mycotoxin (Group 3), we assumed a correlation between them, but we could not prove this because we could not find significant and biologically interpretable correlations.

The significantly higher ZEA concentrations in fetuses, particularly when we could not detect ZEA in the hinds, reflect the possible accumulation of ZEA in the fetus. The mycotoxin concentration showed an increasing trend with the age of the fetuses—there were slight increases in all cases, but only DON was statistically significant, indicating the same trend for the other mycotoxins. This has drawn attention to the threat of mycotoxin intake during pregnancy. Further studies are needed to determine what kinds of reproductive problems can be related to mycotoxins, especially ZEA and DON, and could decrease the number of surviving calves.

## 5. Conclusions

Based on our results, we can conclude that ZEA, AF B1, DON, T2/HT2-toxin, and FB1 are present in many pregnant fallow deer hinds and in the fetuses, even in the best-managed populations. High mycotoxin concentrations and the simultaneous presence of different mycotoxins, even at low concentrations (the multi-mycotoxin effect), can have a negative influence on fetal development and neonatal mortality. The possible accumulation of several mycotoxins in the fetus can cause additional harm because even the low concentration of mycotoxins in the food of the mother could threaten fetal development later. Further studies on reproductive success are required to validate these assumptions.

The high individual variability versus the lower difference among the game management units reflects the highly variable access to mycotoxin sources in space and time. The type, spatial distribution, and temporal change in mycotoxin sources are important information for risk-decreasing management. The spatial scale of research must be adjusted to the movement and habitat use of the species. As these mycotoxin sources are accessible not only to fallow deer but also to other wild animal populations, the negative influence of mycotoxins can be more general at the ecosystem level.

## Figures and Tables

**Figure 1 animals-14-01039-f001:**
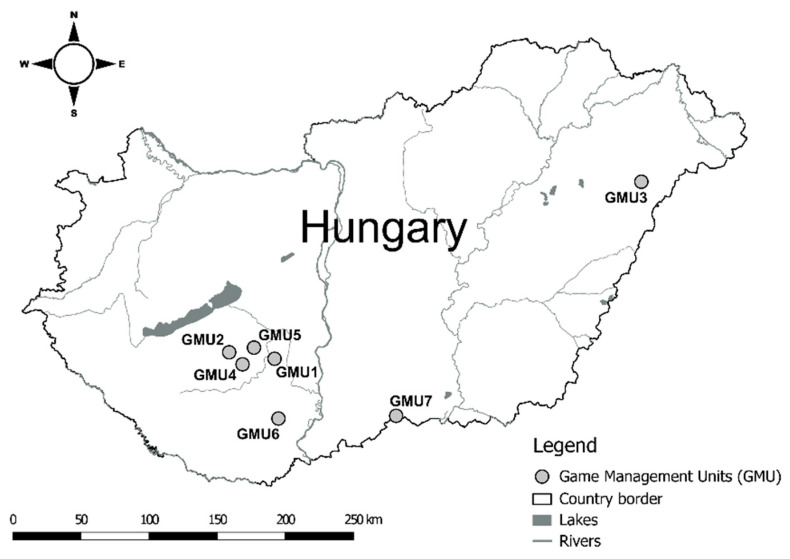
The geographical location of the investigated Game Management Units.

**Figure 2 animals-14-01039-f002:**
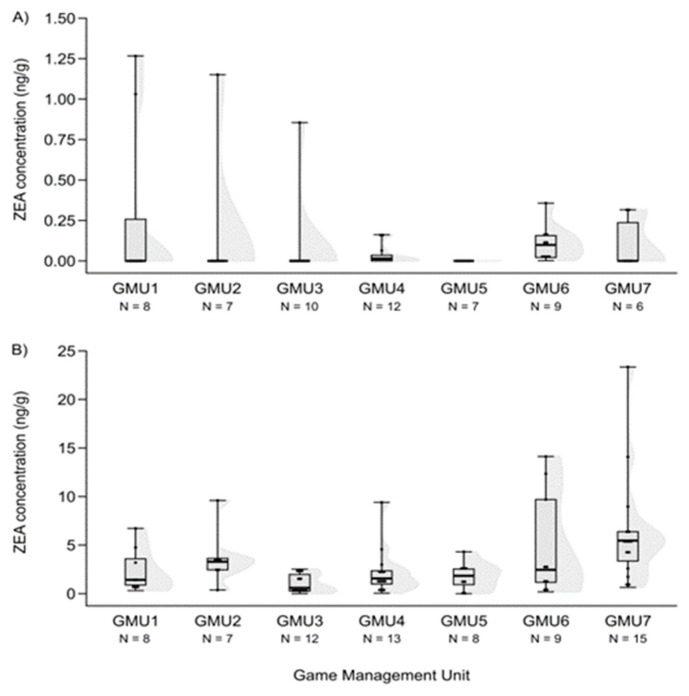
ZEA concentrations in the liver of fallow deer hinds (**A**) and fetuses (**B**) in the sampling areas. (Shaded areas represent distribution).

**Figure 3 animals-14-01039-f003:**
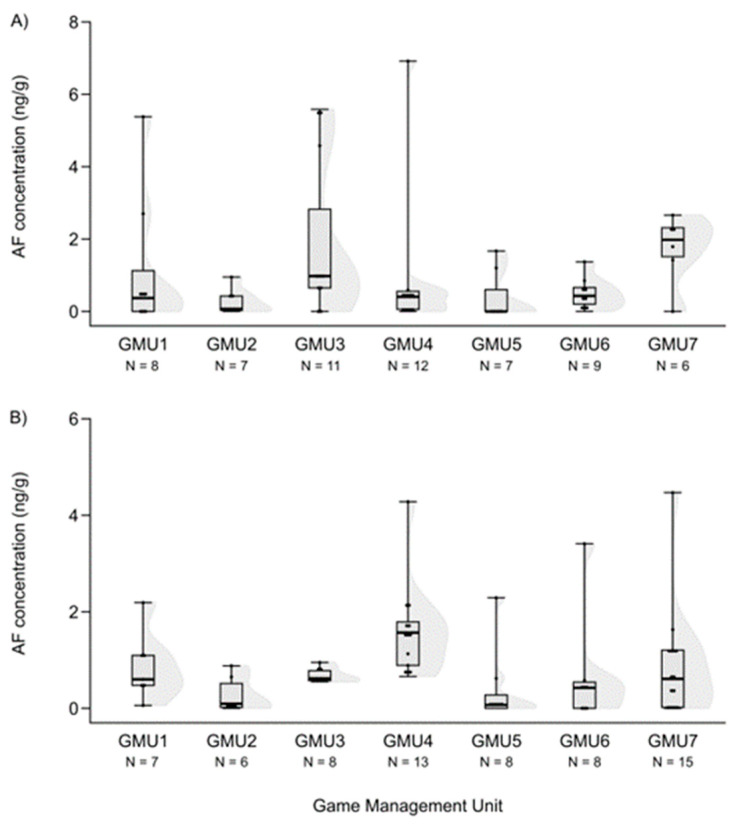
AF concentrations in the liver of fallow deer hinds (**A**) and fetuses (**B**) in the sampling areas. (Shaded areas represent distribution).

**Figure 4 animals-14-01039-f004:**
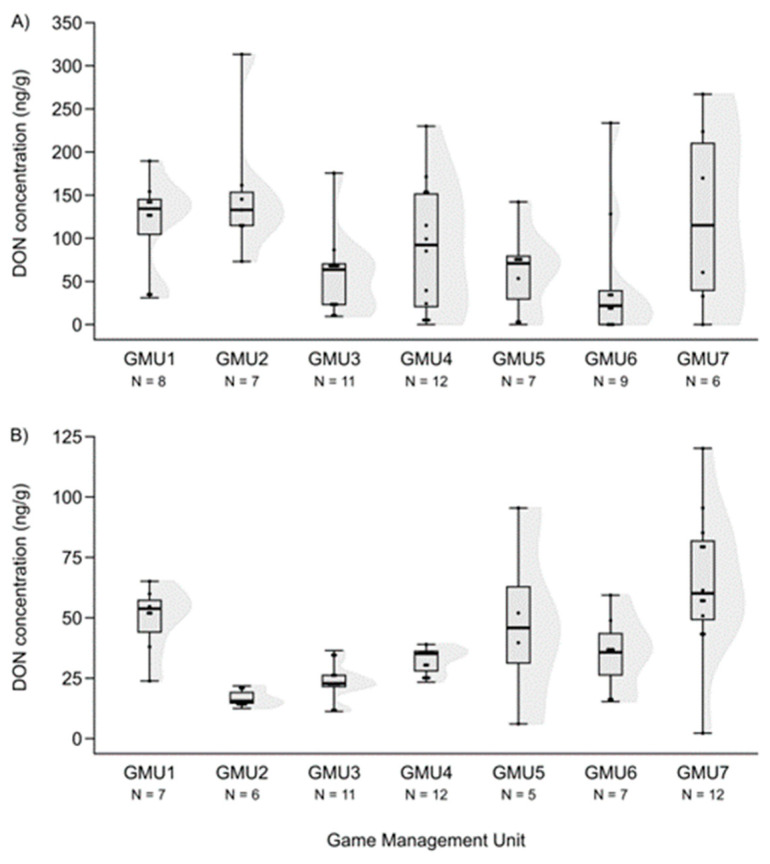
DON concentrations in the liver of fallow deer hinds (**A**) and fetuses (**B**) in the sampling areas. (Shaded areas represent distribution).

**Figure 5 animals-14-01039-f005:**
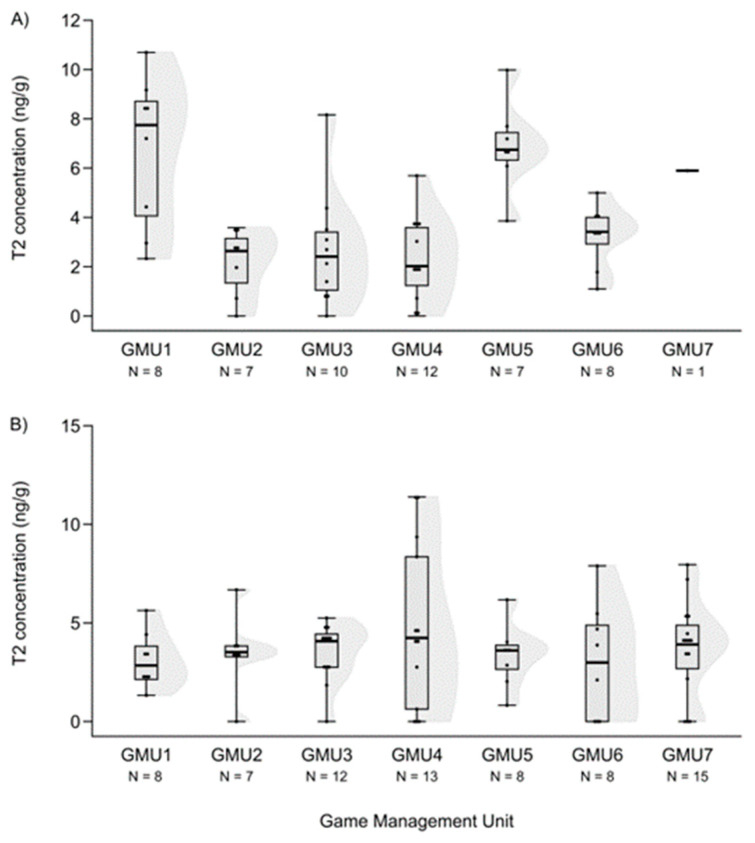
T2-toxin concentrations in the liver of fallow deer hinds (**A**) and fetuses (**B**) in the sampling areas. (Shaded areas represent distribution).

**Figure 6 animals-14-01039-f006:**
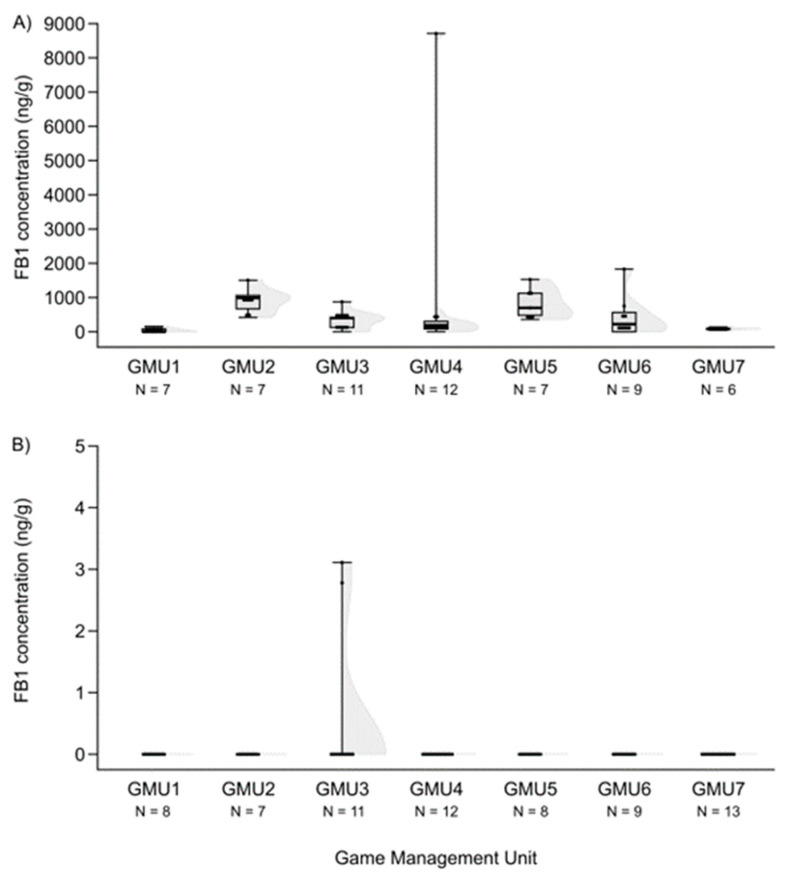
FB1 concentrations in the liver of fallow deer hinds (**A**) and fetuses (**B**) in the sampling areas. (Shaded areas represent distribution).

**Figure 7 animals-14-01039-f007:**
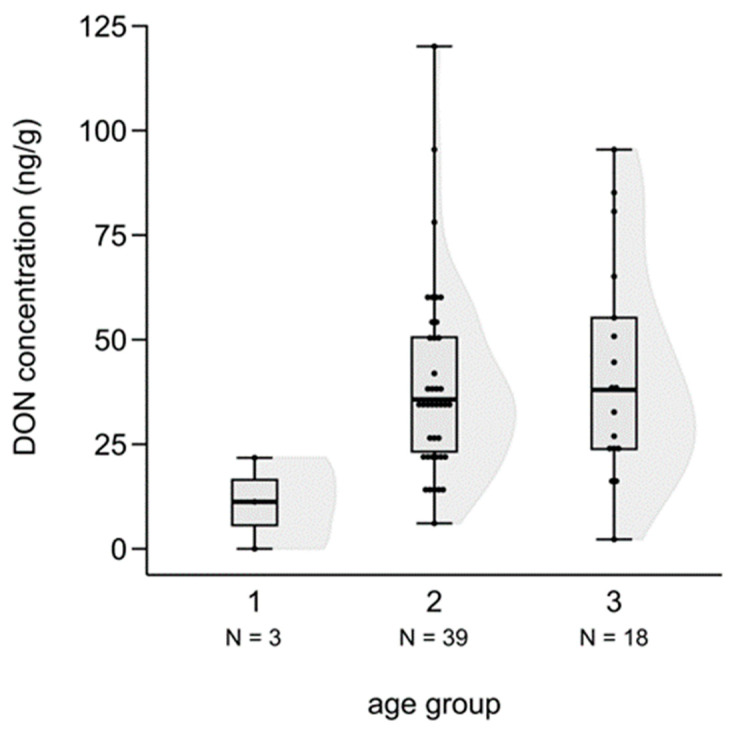
DON concentrations in fallow deer fetus livers according to age groups (1: one-month-old, 2: two-months-old, 3: three-months old). (Shaded areas represent distribution).

**Table 1 animals-14-01039-t001:** Mycotoxin concentrations (ng/g) in the liver of fallow deer hinds.

Mycotoxin	Valid N	Median	Minimum	Maximum	Number of “0” Concentration
AF	60	0.475	0	6.918	18
ZEA	59	0.000	0	1.267	35
DON	60	72.740	0	313.240	6
T2-toxin	53	3.426	0	16.850	2
FB1	59	256.987	0	8711.522	10

AF was present in 82%, ZEA in 96%, DON in 98%, T2-toxin in 85%, and FB1 in 3% of fetuses. Individual differences were also high among the fetuses (Table 2). We always found individuals with “0” concentration, but the proportion of “0” was more moderate in the cases of ZEA, AF, and DON, whereas it was higher in the cases of T2-toxin and especially FB1 than in hinds. For the three mycotoxins (AF, DON, and T2-toxin), concentrations were in a similar order of magnitude in hinds and fetuses, but the maximum value of ZEA was detected in the fetus and that of FB1 in the hinds.

**Table 2 animals-14-01039-t002:** The mycotoxin concentrations (ng/g) in the liver of fallow deer fetuses.

Mycotoxin	Valid N	Median	Minimum	Maximum	Number of “0” Concentration
AF	65	0.610	0	4.470	12
ZEA	72	2.217	0	23.343	3
DON	60	35.685	0	120.120	1
T2-toxin	71	3.810	0	11.392	11
FB1	68	0.000	0	3.110	66

**Table 3 animals-14-01039-t003:** Comparison of the mycotoxin levels (ng/g) between fallow deer hinds and fetuses in sampling sites (Mann–Whitney U test).

Mycotoxin	ZEA		AF		DON		T2-Toxin	
Sampling Area	U	Z	U	Z	U	Z	U	Z
GMU1	6	2.678 **	20	n.s.	11	n.s.	10	−2.258 *
GMU2	-	-	20	n.s.	0	−2.929 **	11.5	n.s.
GMU3	16	2.868 **	24	n.s.	37	n.s.	38.5	n.s.
GMU4	3	4.052 ***	13	3.508 ***	47	n.s.	63.5	n.s.
GMU5	4	2.720 **	24	n.s.	12.5	n.s.	3	−2.835 **
GMU6	1	3.444 ***	30.5	n.s.	24	n.s.	30	n.s.
GMU7	0	3.464 ***	19.5	n.s.	29	n.s.	-	-

* *p* < 0.05, ** *p* < 0.01, *** *p* < 0.001; n.s. = non-significant; - cannot be analyzed.

## Data Availability

Basic data in Access database (accessed on 1 October 2020) and details of statistical analysis can be found at the University of Pécs, Faculty of Sciences, Institute of Biology, (https://www.ttk.pte.hu/karunkrol/intezetek/biologiai-intezet) at Győző Horváth (hgypte@gamma.ttk.pte.hu) and László Szemethy (szemethy.laszlo@pte.hu).

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
