# Peer review of "First Results on the Presence of Mycotoxins in the Liver of Pregnant Fallow Deer (Dama dama) Hinds and Fetuses"

_animals, 2024, doi:10.3390/ani14071039_

Round 1

Reviewer 1 Report (Previous Reviewer 2)

Comments and Suggestions for Authors

Dear Authors,

The aim of the study is interesting. The form of the representation is poor. Too much introduction, but in the discussion of the results there was not much extrapolation and interaction of the results between the study groups. No attempt has been made to analyse in depth the correlation between mycotoxin levels in mothers and their foetuses from the point of view of, or rather the fact that they are ruminants. After all, these are organisms that cope in a very specific way (with the participation of the rumen and gut microbiota) in the process of mycotoxin biotransformation. In addition, the liver and the prehepatic vascular system with erythrocytes are involved in this process. Among the mycotoxins studied, there is one (FB1) that does not undergo rumen biotransformation and the absorption process is very slow. There are more observations of this kind.

The writing is also sloppy, e.g. abbreviations of mycotoxin names are introduced once (when the name mycotoxin is used for the first time). You should not be "reminded" of the abbreviations on each page of the text several times. The names of mycotoxins should always be unambiguous, e.g. what is the difference between T2 and T2 toxin – what is it called.

Author Response

Please find the answers in the file attached.

Reviewer 2 Report (New Reviewer)

Comments and Suggestions for Authors

Line number

Text

Comment

20

We supposed mycotoxin contamination to be one of the possible causes because it is known to be dangerous even at low toxin levels, especially for young animals

Frame as a hypothesis or state we suspect that….

27

The individual variability of mycotoxin levels was extremely high, but the spatial differences were moderate. We found accumulation of ZEA and DON in the foetuses.

How is moderate differences defined. Was this statistically significant. How is accumulation defined (versus stating we detected ZEA and DON in the fetuses).

72

As the influence of climate change is stronger in the Carpathian basin, mold fungus occurrence and mycotoxin contamination have become increasingly serious problem in Hungary [9].

Why is the impact greater in the basin? Basis for this conclusion? Not clear whether there is incidence data to support the conclusion.

74

The prevalence of Fusarium toxins is increasing, and new mold species like Aspergillus flavus and new mycotoxins, especially Aflatoxins have emerged in the last fifteen years.

Similar question  - is there prevalence data, if yes cite source. As written this implies that aflatoxins represent a new class of mycotoxin – they have been known for decades.

95

ZEA can pair up with alpha and beta estrogen receptors (ER) and can interfere with the endocrine system [18].

Use appropriate terminology – e.g., binding, agonist/antagonist actions.

101

In lower levels, ZEA can cause endometrial hyperplasia and therefore it elevates the risk of endometrial cancer [20, 21].

Identify species (humans?)

117

Although the roles of Akt and MAPKs have been elucidated in porcine endometrium, the potential effects of DON on placental function and embryogenesis have not been adequately evaluated [26].

Define abbreviations

137-144

Not needed

174-184

Data source?

179

The game management was similar in the sampling areas, and the fallow deer was the dominant species.

Dominant game species? Dominant cervid?

227

Before the ELISA assays, we compared 10-10 liver samples with HPLC/FLD using our validation methods [5].

10-10?

235

In each game management unit, we used Kruskal-Wallis ANOVA (followed by Dunn’s post hoc test for multiple comparisons) to test the differences in the mycotoxin levels of hinds and foetuses, and the Mann-Whitney test was used to examine the difference between the toxin level of the hinds and foetuses. We also used the Kruskal-Wallis test to compare the mycotoxin concentrations between the different age groups of foetuses [39].

The data suggests that you have outliers – data should be evaluated for outliers using an appropriate statistical test. Were assumptions of the various tests met? What statistical tests were used to evaluate these assumptions? It would also be of interest to evaluate if maternal and fetal concentrations were correlated (e.g., using regression methods)

Tables 1 and 2

Mycotoxin concentrations (ng/g) in the liver of fallow deer hinds/fetuses. What was the LOD for the different assays? What does “Valid N” mean – may be more appropriate to remove the term valid here. It remains unclear to me why sample numbers for some mycotoxins are higher with fetuses versus the adult deer – this is not accounted by twin fetuses.

Figures

Data varies quite a bit between areas and mycotoxins – there does not appear to be one area with a higher likelihood of higher liver mycotoxin concentrations. As noted earlier – I don’t full trust some of these analyses since apparent outliers have not been critically evaluated.

Section 3.3

It is unclear whether the statistical analysis considered paired analyses between dams and fetuses.

Table 3

U and Z should be defined – will require rewording of the Table legend.

Figure 7

Define age groups rather than use a numerical grouping (1,2,3)

368

For cattle, for example, targeting the dietary levels of toxins and endocrine-disrupting metabolites is vital to assessing the risks for impacts on health, reproduction, and production [32, 33, 34].

What is meant by targeting?

379

Hungarian game managers reported reproductive problems…

These problems remain poorly defined.

Discussion

Can be shortened. You should discuss limitations of the study (e.g., prevalence survey only – no data regarding adverse outcomes)

 In summary my main concerns are: (a) no data has been provided to support the conclusion that reproductive anomalies have increased in this deer species; (b) even if some data regarding reproductive outcomes is available the current data does not provide any data regarding temporal changes in mycotoxin concentrations in deer; (c) statistical methods are worrisome since there appears to be significant outliers in the data that have not been evaluated using an appropriate test of outliers; (d) it is also unclear how zero values were handled in the statistical analysis - e.g., some investigators will use the LOD as a zero value; (e) no outcome data is reported; (f) the term accumulates in the fetus is poorly defined. Accumulation may suggest that concentrations seen in the fetus are > than those seen in the dam - it is unclear whether a rigorous statistical analysis was performed; and finally (g) it's unclear why different regions were evaluated since these subanalyses do not provide useful data regarding sources of mycotoxins, exposure, or outcome. Some but not all of these concerns could be addressed by modifying the text while others will require thoughtful reconsideration of the statistical approaches used.

Comments on the Quality of English Language

A fair amount of editing, especially the Discussion, will be required.

Author Response

Please find the answers in the file attached.

Reviewer 3 Report (New Reviewer)

Comments and Suggestions for Authors

The idea to study the potentialy toxiticy of mycotoxin on embrios and foetus is very intersting. The research is well conducted even is there is an area of sampling very far from the others and maybe to exclude. Unfortunately the results are not encouraging so it will be usefull to search another reason of problems at birth. Maybe another article in the future! Something to control on the English form: sometimes, it seems Google translate. Intersesting for readers (this is only the reason of minor revision, nothing else).

Comments on the Quality of English Language

Some paragraph to improve: it seems Google translate!

Author Response

Please find the answers in the file attached.

Reviewer 4 Report (New Reviewer)

Comments and Suggestions for Authors

The work is well structured with good data analysis.

The sampling chosen needs to be clarified because it would have been interesting to have a comparison with non-pregnant adult females, probably allowing us to better explain the hypotheses relating to the distribution of mycotoxins in the female/foetus groups being analysed.

Ll 72-74: It is better to include a more recent reference regarding the influence that climate change has on the presence of mold fungus and mycotoxins.

L 188: Explain here and in the introduction why it was not chosen to also sample non-pregnant adult females. If all sampled subjects were pregnant females then it is necessary to modify the conclusions which state that the presence of mycotoxins in foods can directly affect pregnancy, in fact it is presumable that the effect only exists in terms of foetal development and/or neonatal mortality.

L 240: Specify what is meant by “different age groups of feet” and which method was used for the classification.

Ll 281-284: transfer after figure 4.

Author Response

Please find the answers in the file attached.

Reviewer 5 Report (New Reviewer)

Comments and Suggestions for Authors

The study collected 72 pregnant fallow deer from seven areas and extracted the concentrations of various mycotoxins in maternal and fetal livers, aiming to investigate the spatial patterns of mycotoxin occurrence and the relationship between maternal and fetal mycotoxin levels. Overall, the article is well structured, methodologically rigorous, and provides valuable data that are important for understanding the distribution of mycotoxins in wildlife and their health effects. In terms of details, I suggest that the authors could correct the following issues to further improve the quality of the article:

1. Most of the preface of the article is devoted to the negative effects of different types of mycotoxins on animal reproduction, which should have been shorter, and it is recommended that the authors emphasize the importance of the study more in the preface.

2. I noticed that the introductory section of the article seemed to lack a clear description of the research hypothesis. Including a description of the hypothesis would not only clarify the intended goals and questions of the study for the reader, but it would also enhance the orientation and relevance of the article. It is recommended that the authors add the hypothesis.

3. In the results the authors used the median to express the concentration of mycotoxins, I noticed that the median of ZEA in Table 1 is 0. I think that the median in the present data may not be able to adequately express the accuracy of this dataset, and I suggest that the authors may consider expressing it in a different parameter.

4. In result 3.2, it is recommended that the authors simplify the parameters that follow each result, keeping only the p-values, so that the variability between the different groups can be more clearly understood when reading.

5. The contents of Figures 2 to 6 are consistent, and it is suggested that the authors may consider combining these figures into 1 or 2 large ones so that the pictures can be more intuitive about the differences in concentrations of different regions and molds.

6. In result 3.3, when the authors analyzed the difference between doe and fetus, the authors took the regional factor into consideration, and I suggest that the authors could have excluded the regional factor first and analyzed the difference in mycobacterial concentration between doe and fetus in general.

7. Are there differences in the concentration of different species of molds in the same area, and is there a correlation between different species of molds? It is suggested that the authors might consider performing some analysis to address this question.

8. In the discussion the authors focus on the results as a whole, and it is suggested that the authors could have discussed the results in more detail, such as why some molds may vary between certain areas and not others.

I hope that these suggestions I have made will be of some help to authors.

Author Response

Please find the answers in the file attached.

Round 2

Reviewer 2 Report (New Reviewer)

Comments and Suggestions for Authors

Please add information about using "0" as a lower value for the non-detects (versus using the LOD as truncated data).

Table 3 - legend clearly indicate the statistical test used.

I appreciate the comments regarding outliers - it's not uncommon to evaluate outliers prior to running an ANOVA (or similar analysis) since this can address the finding that the data not normally distributed. Likewise afterwards some data transformation methods can also be applied and attemoted prior to using nonparametric approaches.

Comments on the Quality of English Language

Some light editing will be required

Author Response

Answers to Reviewer2  round2

Article: „First results of mycotoxin exposition of pregnant Fallow deer (Dama dama) hinds and foetuses” by István Lakatos 1, Bianka Babarczi 2, Zsófia Molnár 2, Arnold Tóth 2, Gabriella Skoda 2, GyÅ‘zÅ‘ F. Horváth 3, Adrienn Horváth 3, Dániel Tóth 3, Farkas Sükösd 4, László Szemethy 3*† and Zsuzsanna SzÅ‘ke 2†

Thanking the review and useful suggestion please find our answer here.

“Please add information about using "0" as a lower value for the non-detects (versus using the LOD as truncated data).”

We added the sentence “(The values of samples in which the given mycotoxin was not detected, or the concentrations were below LOD were considered zero.)” to clarify data treatment.

“Table 3 - legend clearly indicate the statistical test used.”

We added “(Mann-Whitney U test)” to the legend.

“I appreciate the comments regarding outliers - it's not uncommon to evaluate outliers prior to running an ANOVA (or similar analysis) since this can address the finding that the data not normally distributed. Likewise afterwards some data transformation methods can also be applied and attemoted prior to using nonparametric approaches.”

Yes indeed, the data transformation could be useful if we want to focus on statistical tests and we suppose that the outliers are happen randomly. We considered several transformations, but we rejected to use those, because i) all of those have some mathematical problems (like log transformation with the numerous “0” values); ii) the individual differences were important in our study. The proportion of “0” or near “0” values and the extreme high mycotoxin concentrations reflect to the variability inside a local population and warn us to the weaknesses of general management of the problems (reproduction and food safety). That is why we decided to present the data distribution in the figures. We can apply the results on the variability in the planning of our future research.

GödöllÅ‘, 17.03.2024

László Szemethy

This manuscript is a resubmission of an earlier submission. The following is a list of the peer review reports and author responses from that submission.

Round 1

Reviewer 1 Report

Comments and Suggestions for Authors

This is an interesting article on the presence of mycotoxins in hinds and their foetuses, and may clarify the reproductive alterations observed in these game animals. However, it presents several issues that must be clarified and modified before its final acceptance.

Abstract section:

-I consider it is important to include more information, such as the percentage of appearance of each mycotoxin as indicated in lines…

Introduction section:

-There is no description of the objectives that should appear at the end of the introduction section.

Sampling section:

-Lines 177-182: There is an irregularity in the number of deer sampled. It is indicated in the article that 70 hinds were analysed but that in 11 of them the liver samples were missing or not suitable for mycotoxin analysis due to large liver fluke infection. Therefore, in Table 1 the valid number of samples cannot be greater than 59, and an "N" of 60 appears in AF and DON and 62 in FB1. Clarify this.

-Furthermore, for a better understanding of the results of the statistical study relating to the comparison between sampling zones (Figures 2 to 6), the number of samples taken in each zone should be indicated in the text or in a table.

-Line 179: correct the sentence “The sample sizes my differ at different…”

-Lines 191, 192 and 212: Why the number is repeated in 0.5-0.5g, 0.5-0.5ml and 10-10 liver samples?

-Lines 196-197: for a better understanding to readers consider to change the sentence to “After incubation and extraction with a 70% methanol: water mixture (3:1) (v/v), the mixture was shaken for 15 min at room temperature (RT) on an orbital shaker.” In line 198 “RT” is supposed to mean “room temperature”.

-Lines 201-202 and 205-206: unify in one sentence “For the total aflatoxin and deoxynivalenol (DON) tests Toxi-Watch (Soft Flow Ltd., Pécs) immunoassay kits were used, according to the manufacturer's instructions”.

-Lines 207-208: improve the sentence, avoid repetitions.

-Line 226: indicate the specific conditions of each of the 3 groups of the foetuses.

Results section:

-Lines 248-250: It is stated that “The mean and the median mycotoxin concentrations were in the similar order of magnitude in hinds and foetuses, but the maximum was detected in a foetus at ZEA and in a hind at FB1.”. However, no means are included in Tables 1 and 2. For a better understanding this sentence could be modified as “…. order of magnitude for AF, DON and T2-toxin in hinds and foetuses…”

-Lines 284-295: This paragraph describe the statistical analysis of foetuses mycotoxin levels between sampling sites, not between hinds and foetuses. So, it must be moved from subsection “3.3” to subsection “3.2. Spatial pattern of mycotoxin concentrations in the hinds and foetuses”

-Lines 358 to 361: This sentence is a conclusion, so it should be moved to the conclusions section.

-Lines 374-377: The sentence is incomplete, correct to “In the hunting season… game managers are also available.”

-Lines 387-388: The sentence is incomplete “There were apparent differences among the by mycotoxins in the number of samples in this group.”

References section:

Authors should review references to maintain consistency in their writing (see “Instructions for Authors” of the journal). The same criteria must be followed in all references.

-Title of the journals: some of them appear with the full title and others abbreviated, and those abbreviates some appears with full stop - J. Chromatogr. B.- and other ones without full stop - Br Med J (Clin Res Ed)-).

-The title of references 1, 5, 9, 11, 14, 15, 23, 25, 31, 36 and 37 have the first letter of each word capitalized, while in most titles all words (except the first) are in lower case. Put them in lower case.

- The DOI number is written in different ways: doi: 10.1179/027249302125000094  and https://doi.org/10.3390/toxins14090634 and DOI: 10.1016/j.tvjl.2009.03.002 )

General issues:

A period should be put at the end of the sentence in lines 67, 74, 97, 120……

Author Response

Thank you for the reviewing our article. Considering recommendations, we have revised the paper. Please find the answers attached.

Answers to Reviewer1

Article: „First results of mycotoxin exposition of pregnant Fallow deer (Dama dama) hinds and foetuses” by István Lakatos 1, Bianka Babarczi 2, Zsófia Molnár 2, Arnold Tóth 2, Gabriella Skoda 2, GyÅ‘zÅ‘ F. Horváth 3, Adrienn Horváth 3, Dániel Tóth 3, Farkas Sükösd 4, László Szemethy 3*† and Zsuzsanna SzÅ‘ke 2†

Thanking the review and useful suggestion please find our answer here.

Reviewers suggested to change the title. We have replaced the “exposition” with presence and inserted “liver” to make the title more correct and informative.  The new title is “First results on the presence of mycotoxins in the liver of pregnant fallow deer (Dama dama) hinds and foetuses”.

Abstract section:

“I consider it is important to include more information, such as the percentage of appearance of each mycotoxin as indicated in lines…”

We have improved the summary and abstract by insertion the most important results.

“Introduction section:

-There is no description of the objectives that should appear at the end of the introduction section.”

We have completed the introduction with the goals of the study.

Sampling section:

- “Lines 177-182: There is an irregularity in the number of deer sampled. It is indicated in the article that 70 hinds were analysed but that in 11 of them the liver samples were missing or not suitable for mycotoxin analysis due to large liver fluke infection. Therefore, in Table 1 the valid number of samples cannot be greater than 59, and an "N" of 60 appears in AF and DON and 62 in FB1. Clarify this.”

Yes, it was not correct, because only eight hind liver samples were missing or not suitable for mycotoxin analysis due to large American liver fluke. In some other cases small quantity could be taken only, especially from very small foetuses, that was not enough for the analysis every toxin. The real sample sizes are presented in Table 1 and 2. We have clarified it in the text.

- “Furthermore, for a better understanding of the results of the statistical study relating to the comparison between sampling zones (Figures 2 to 6), the number of samples taken in each zone should be indicated in the text or in a table.

We have added the sample sizes to the figures.

- “Line 179: correct the sentence “The sample sizes my differ at different…””

We have corrected the sentence.

- “Lines 191, 192 and 212: Why the number is repeated in 0.5-0.5g, 0.5-0.5ml and 10-10 liver samples?”

It was a misdrawing. We have clarified in the text.

- “Lines 196-197: for a better understanding to readers consider to change the sentence to “After incubation and extraction with a 70% methanol: water mixture (3:1) (v/v), the mixture was shaken for 15 min at room temperature (RT) on an orbital shaker.” In line 198 “RT” is supposed to mean “room temperature”.”

We have clarified in the text.

- “Lines 201-202 and 205-206: unify in one sentence “For the total aflatoxin and deoxynivalenol (DON) tests Toxi-Watch (Soft Flow Ltd., Pécs) immunoassay kits were used, according to the manufacturer's instructions”.”

Thank you for the suggestion. We have done it.

- “Lines 207-208: improve the sentence, avoid repetitions.”

We have done it.

- “Line 226: indicate the specific conditions of each of the 3 groups of the foetuses.”

We made the age determination method clearer in the 2.2 Samling chapter. We added two citations, both studies were done on fallow deer in Hungary. The correlations between the age and the bodyweight and the age and the body length and the variability are determined in these publications. Our classification is based on these results.

Results section:

- “Lines 248-250: It is stated that “The mean and the median mycotoxin concentrations were in the similar order of magnitude in hinds and foetuses, but the maximum was detected in a foetus at ZEA and in a hind at FB1.”. However, no means are included in Tables 1 and 2. For a better understanding this sentence could be modified as “…. order of magnitude for AF, DON and T2-toxin in hinds and foetuses…””

Mention of mean remained from an earlier version. As the distribution of data is not normal it was a mistake to calculate the mean, that is why we eliminated it from Table 1 an2. We have rewritten the sentence.

- “Lines 284-295: This paragraph describe the statistical analysis of foetuses mycotoxin levels between sampling sites, not between hinds and foetuses. So, it must be moved from subsection “3.3” to subsection “3.2. Spatial pattern of mycotoxin concentrations in the hinds and foetuses””

We translocated the paragraph.

- “Lines 358 to 361: This sentence is a conclusion, so it should be moved to the conclusions section.”

We have translocated the sentence.

-“Lines 374-377: The sentence is incomplete, correct to “In the hunting season… game managers are also available.’”

We have rephrased the sentence.

-“Lines 387-388: The sentence is incomplete “There were apparent differences among the by mycotoxins in the number of samples in this group.””

We have clarified it.

References section:

“Authors should review references to maintain consistency in their writing (see “Instructions for Authors” of the journal). The same criteria must be followed in all references.

-Title of the journals: some of them appear with the full title and others abbreviated, and those abbreviates some appears with full stop - J. Chromatogr. B.- and other ones without full stop - Br Med J (Clin Res Ed)-).

-The title of references 1, 5, 9, 11, 14, 15, 23, 25, 31, 36 and 37 have the first letter of each word capitalized, while in most titles all words (except the first) are in lower case. Put them in lower case.

- The DOI number is written in different ways: doi: 10.1179/027249302125000094  and https://doi.org/10.3390/toxins14090634 and DOI: 10.1016/j.tvjl.2009.03.002 )”

We put the references in the required format.

General issues:

“A period should be put at the end of the sentence in lines 67, 74, 97, 120……”

We have corrected.

Thank you for your comment once again.

Reviewer 2 Report

Comments and Suggestions for Authors

The paper entitled "First results of mycotoxin exposition of pregnant Fallow deer (Dama dama) hinds and foetuses" presents the results of research on the level of selected mycotoxins in the liver of pregnant fallow deer and their foetuses

However, I have a few comments and I ask you to refer to them by explaining the situation by supplementing the paper:

1.      The INTRODUCTION and the DISCUSSION chapter are written very chaotically and poorly substantively. Both chapters should be deepened and updated (no references to works from the last two or three years);

-        Secondly, it cannot be the case that the authors analyze the presence of mycotoxins in specific tissues of ruminants such as fallow deer and do not take into account the fact that the biotransformation of mycotoxins in ruminants is very different;

-        Thirdly, in my opinion, it is impossible to compare the course of biotransformation processes in ruminants with similar processes in pigs or poultry;

2.      Row 54 – should be "selected mycotoxins";

3.      Row 55 – The term "all mycotoxins" is a wish, not a certainty, because every now and then we discover the presence of new mycotoxins and their metabolites. Therefore, this passage should be reworded;

4.      Lines 56/57 – The statement 'detection of both exposure levels' is inherently false. In the case of wild animals, we do not know the set of mycotoxin-contaminated flora they take up and the duration of exposure;

5.      Row 63 – Mycotoxins do not spread, but favorable climatic conditions are created for the environment to be infected by incoming mold fungi producing specific mycotoxins;

6.      Verse 66 – It should be "mold fungus";

7.      Lines 69/71 – The sentence is untrue from a mycological point of view;

8.      Paragraph 110 to 122 lines – Throughout this paragraph, the authors document their suggestions with rather archaic literature. Nowadays, when using the Internet and on-line bibliography, one should support oneself with data from the literature on the subject (and by the way – what does poultry have to do with fallow deer) and from the last two or three years;

9.      Verse 118 – This is a very inaccurate statement. It should be written that in values equal to or below NOAEL, i.e. in MABEL values;

10.  Between lines 153-154 – the purpose of the research is missing;

11.  Verse 176 – Why were pregnant hinds shot?

12.  Row 179 – The relationship between the size of the samples taken and the number of mycotoxins determined is unclear;

13.  Entire chapter DISCUSSION – In this chapter, there is no analysis or extrapolation of the obtained results in relation to the processes of mycotoxin biotransformation at the level of the rumen of fallow deer and further sections of the gastrointestinal tract. The interaction of the gut microbiota in the absorption and biotransformation of the gut microbiota has not been addressed. The presence and influence of metabolites of the studied stem mycotoxins was not discussed at all. In fact, this whole chapter is very weak in terms of content to draw any conclusions. I therefore propose to reword it. I propose to withdraw all literature relating to poultry and pigs;

14.  Entire chapter CONCLUSIONS - These are not conclusions, this is a summary. Secondly, it would be better to give up the literature, it should be the result of your reflections and not loose observations on the subject.

Author Response

Thank you for the reviewing our article. Considering recommendations, we have revised the paper. Please find our answers attached.

Answers to Reviewer2

Article: „First results of mycotoxin exposition of pregnant Fallow deer (Dama dama) hinds and foetuses” by István Lakatos 1, Bianka Babarczi 2, Zsófia Molnár 2, Arnold Tóth 2, Gabriella Skoda 2, GyÅ‘zÅ‘ F. Horváth 3, Adrienn Horváth 3, Dániel Tóth 3, Farkas Sükösd 4, László Szemethy 3*† and Zsuzsanna SzÅ‘ke 2†

Thanking the review and useful suggestion please find our answer here.

Reviewers suggested to change the title. We have replaced the “exposition” with presence and inserted “liver” to make the title more correct and informative.  The new title is “First results on the presence of mycotoxins in the liver of pregnant fallow deer (Dama dama) hinds and foetuses”.

“1.The INTRODUCTION and the DISCUSSION chapter are written very chaotically and poorly substantively. Both chapters should be deepened and updated (no references to works from the last two or three years);”

We restructured the Introduction, and we added several new citations. Although we want to note, that as the readers of this paper cold be not only the mycotoxin expert but rather wildlife biologists, we think to be useful to give some general information about the known influences of the mycotoxins. We also remark that the diet and digestion of wild ruminants especially deer species differ from the domestic ones, as it is proved in many papers including our previous studies too, so those are applicable for games very carefully only. 

We have also restructured and rewritten the discussion.   

- “Secondly, it cannot be the case that the authors analyse the presence of mycotoxins in specific tissues of ruminants such as fallow deer and do not take into account the fact that the biotransformation of mycotoxins in ruminants is very different;”

This and the next objection of the Reviewer could derive from a misunderstanding. Firstly, we know that the biotransformation of the mycotoxins is a very complex process and there are significant differences among the different species with different digestion physiology. We also know that the biotransformation of mycotoxins in deer species is known poorly. That is why we had to analyse the mycotoxin metabolites which were described in domestic ruminants. We did not analyse the biotransformation of the mycotoxins in the fallow deer. It is hardly feasible among natural conditions on wild animals. We did not analyse the diet and the mycotoxin levels of different consumed plants because it would exceed the capacity of this paper. We simple intended to describe the first results for the species. One source of the misunderstanding could be the misleading use of the “exposition” in the title. We have rephrased the title to make more clear the objective of the paper. We analysed the liver only, because based on the pilot studies the mycotoxin concentrations in this organ seemed to be the most informative. The new title is: “First results on the presence of mycotoxins in the liver of pregnant fallow deer (Dama dama) hinds and foetuses”.

- “Thirdly, in my opinion, it is impossible to compare the course of biotransformation processes in ruminants with similar processes in pigs or poultry;”

We do not understand this objection clearly. As we wrote above, we do not study the biotransformation in this paper. On the other hand, we give some basic information about the possible harms of mycotoxins in different mammal focusing on ruminant if it is possible. We had to do this, because as we wrote there are very few studies available on wild ruminants. The only citation is 21. Bodó et al. “Possible applications of poultry immunoglobulins focusing on mycotoxin environmental loads and human influence” where poultry is mentioned but only using IgY produced by poultry for analysis. We have deleted the citation. We had to mention laboratory animals several times and saws once to present the possible harmful influence of mycotoxins for the pregnancy and the foetus. We added some more citations on ruminants especially cattle to strengthen this part.

  1. “Row 54 – should be "selected mycotoxins";” and 3. Row 55 – The term "all mycotoxins" is a wish, not a certainty, because every now and then we discover the presence of new mycotoxins and their metabolites. Therefore, this passage should be reworded;”

This part talks about an ‘ideal’ parallel detection of the mycotoxins. Of course, it is not possible to analyse all kind of mycotoxins in one step now. The reviewer’s recommendation helps to avoid misunderstanding.  We have changed wording.

  1. “Lines 56/57 – The statement 'detection of both exposure levels' is inherently false. In the case of wild animals, we do not know the set of mycotoxin-contaminated flora they take up and the duration of exposure;”

Yes, it is completely right. We have rewritten this part to make it clear that it is a vision. We think it necessary, because we want to emphasise the need for more intensive studies even for wildlife.

  1. “Row 63 – Mycotoxins do not spread, but favorable climatic conditions are created for the environment to be infected by incoming mold fungi producing specific mycotoxins;”

Yes, it was a wrong drawing. We have rewritten that part.

  1. “Verse 66 – It should be "mold fungus";”

We have rewritten it.

  1. “Lines 69/71 – The sentence is untrue from a mycological point of view;”

Yes, it was misunderstandable. We have rephrased it.

  1. “Paragraph 110 to 122 lines – Throughout this paragraph, the authors document their suggestions with rather archaic literature. Nowadays, when using the Internet and on-line bibliography, one should support oneself with data from the literature on the subject (and by the way – what does poultry have to do with fallow deer) and from the last two or three years;”

See, our reflections at the section1, please. Of course, we made deep literature survey, and we make it continuously. We use different literature observer services like Academia.edu, Scopus, Web of Science etc. One of the results is that there are very few research papers on the wild animals and practically nothing on fallow deer. That is why we started our research work and as it is a first step, we can rely only the general literature.

  1. “Verse 118 – This is a very inaccurate statement. It should be written that in values equal to or below NOAEL, i.e. in MABEL values;”

The no observed adverse effect level (NOAEL) is defined as the highest dose where the effects observed in the treated group do not imply an adverse effect to the subject.

EMA guidance stresses the minimal anticipated biological effect level (MABEL) approach, in which all in vitro and in vivo information will be taken into consideration. The NOAEL- or the MABEL-derived human equivalence dose can be reduced further by applying the safety factor, a number by which the calculated human equivalence dose is divided to increase the assurance that the first dose will not cause toxicity in humans. (Suh et al, 2016)

We think these limits irrelevant to our study because these are about the intake by food or feed. We did not study the intake, because it would be difficult for wild animals living among natural conditions where the diet composition is not known. Determination of the diet and revealing the main mycotoxin sources would be important step in the future studies. The message of our article is that these mycotoxins can be find in most fallow deer hinds and foetuses and to draw the attention to the problem of wild animals living among natural conditions.

On the other hand, these kinds of limitations are designated for the food humans and for the feed of domestic animals. The values differ among countries or international organizations. For example, although the European Union has suggestions for the limits the Hungarian recommendations are stricter. Moreover, these limits are determined for domestic animals only. Tere are no limits recommended for games. As we mentioned above the diet and digestion physiology of wild ruminants, especially deer species differs from the domestic ruminants so it would be difficult to apply the limits to fallow deer directly.  

In this paragraph we simply gave some possible and known harms caused by mycotoxins.

Suh HY, Peck CC, Yu KS, Lee H. Determination of the starting dose in the first-in-human clinical trials with monoclonal antibodies: a systematic review of papers published between 1990 and 2013. Drug Des Devel Ther. 2016 Dec 8;10:4005-4016. doi: 10.2147/DDDT.S121520. PMID: 27994442; PMCID: PMC5153257.

  1. “Between lines 153-154 – the purpose of the research is missing;

We have completed the introduction with the goals of the study.

  1. “Verse 176 – Why were pregnant hinds shot?”

The rutting season is in October the calving is in June in fallow deer populations in Central Europe. The hunting season is from October to February. The population must be regulated by shooting of hinds. Almost all adult hinds are pregnant, and it is not possible to select the pregnant or non-pregnant hinds in the first trimester of pregnancy. The hunter shot the animals during the regular hunts on usual ways. All of the hinds in the bag were eviscerated as soon as possible after shooting and so it was possible to select the pregnant ones. 

  1. “Row 179 – The relationship between the size of the samples taken and the number of mycotoxins determined is unclear;”

Yes, it was not correct, because only eight hind liver samples were missing or not suitable for mycotoxin analysis due to large American liver fluke. In some other cases small quantity could be taken only, especially from very small foetuses, that was not enough for the analysis every toxin. The real sample sizes are presented in Table 1 and 2. We have clarified it in the text.

  1. “Entire chapter DISCUSSION – In this chapter, there is no analysis or extrapolation of the obtained results in relation to the processes of mycotoxin biotransformation at the level of the rumen of fallow deer and further sections of the gastrointestinal tract. The interaction of the gut microbiota in the absorption and biotransformation of the gut microbiota has not been addressed. The presence and influence of metabolites of the studied stem mycotoxins was not discussed at all. In fact, this whole chapter is very weak in terms of content to draw any conclusions. I therefore propose to reword it. I propose to withdraw all literature relating to poultry and pigs;”

We have restructured the Discussion. Please see our arguments above too.

  1. “Entire chapter CONCLUSIONS - These are not conclusions, this is a summary. Secondly, it would be better to give up the literature, it should be the result of your reflections and not loose observations on the subject.”

We have restructured and rephrased the Conclusions.

Thank you for your comment once again.

Round 2

Reviewer 2 Report

Comments and Suggestions for Authors

no comments